# Immune Correlates of Protection from Filovirus Efficacy Studies in Non-Human Primates

**DOI:** 10.3390/vaccines10081338

**Published:** 2022-08-18

**Authors:** Cheryl A. Triplett, Nancy A. Niemuth, Christopher Cirimotich, Gabriel Meister, Mimi Guebre-Xabier, Nita Patel, Mike Massare, Greg Glenn, Gale Smith, Kendra J. Alfson, Yenny Goez-Gazi, Ricardo Carrion

**Affiliations:** 1Battelle, Columbus, OH 43201, USA; 2Novavax Inc., Gaithersburg, MD 20878, USA; 3Texas Biomedical Research Institute, San Antonio, TX 78227, USA

**Keywords:** correlates of protection, PsVNA, ELISA, Marburg virus, Sudan virus, Ebola virus

## Abstract

Non-human primate (NHP) efficacy data for several Ebola virus (EBOV) vaccine candidates exist, but definitive correlates of protection (CoP) have not been demonstrated, although antibodies to the filovirus glycoprotein (GP) antigen and other immunological endpoints have been proposed as potential CoPs. Accordingly, studies that could elucidate biomarker(s) that statistically correlate, whether mechanistically or not, with protection are warranted. The primary objective of this study was to evaluate potential CoP for Novavax EBOV GP vaccine candidate administered at different doses to cynomolgus macaques using the combined data from two separate, related studies containing a total of 44 cynomolgus macaques. Neutralizing antibodies measured by pseudovirion neutralization assay (PsVNA) and anti-GP IgG binding antibodies were evaluated as potential CoP using logistic regression models. The predictive ability of these models was assessed using the area under the receiver operating characteristic (ROC) curve (AUC). Fitted models indicated a statistically significant relationship between survival and log base 10 (log_10_) transformed anti-GP IgG antibodies, with good predictive ability of the model. Neither (log_10_ transformed) PsVNT_50_ nor PsVNT_80_ titers were statistically significant predictors of survival, though predictive ability of both models was good. Predictive ability was not statistically different between any pair of models. Models that included immunization dose in addition to anti-GP IgG antibodies failed to detect statistically significant effects of immunization dose. These results support anti-GP IgG antibodies as a correlate of protection. Total assay variabilities and geometric coefficients of variation (GCVs) based on the study data appeared to be greater for both PsVNA readouts, suggesting the increased assay variability may account for non-significant model results for PsVNA despite the good predictive ability of the models. The statistical approach to evaluating CoP for this EBOV vaccine may prove useful for advancing research for Sudan virus (SUDV) and Marburg virus (MARV) candidate vaccines.

## 1. Introduction

The Ebola virus (EBOV) outbreaks in West Africa, particularly in 2014, have resulted in the rapid advancement of multiple EBOV vaccine candidates for clinical testing. Some of these products were tested during later outbreaks, including an outbreak in the Democratic Republic of Congo in 2018–2020. As a result, two products were approved: one by both the European Medicines Agency (EMA) and the FDA [1,2] and one by the EMA only [3,4].

Although non-human primate (NHP) efficacy data for several EBOV vaccine candidates have existed for years, definitive correlates of protection (CoP) have not been demonstrated. Data generated from more recent studies and studies using well-characterized wild-type challenge virus and discussion associated with the FDA-led Filovirus Immunology Symposium held 12 December 2014 suggest that antibodies to the filovirus glycoprotein (GP) antigen and other immunological endpoints could correlate with protection [5,6]. Further, it is likely that EBOV and other filovirus vaccines will require development and testing in accordance with 21 CFR 314.600 through 314.650 (drugs) or 21 CFR 601.90 through 601.95 (biological products) (commonly termed the FDA animal rule [AR]), or AR-associated concepts, prior to approval. Accordingly, studies that could elucidate biomarker(s) that statistically correlate, whether mechanistically or not, with protection are warranted.

An immune correlate of protection is an immune response measurement that helps predict vaccine efficacy (protection) [7]. In this study, three immune response measurements were evaluated: neutralizing antibodies using the pseudovirion neutralization assay (PsVNA) with two assay readouts (PsVNT_50_, PsVNT_80_) and anti-GP IgG binding antibodies using the anti-GP IgG ELISA with one assay readout (anti-GP IgG concentration). Immune response measurements on Study Day 28, which is 7 days post second immunization and 14 days prior to challenge, were used for analysis in this study. Vaccine efficacy (protection) was measured with respect to animal survival to end of study.

The primary objective of this study was to perform statistical analyses to evaluate potential CoP for the Novavax EBOV GP vaccine candidate with Matrix-M^TM^ adjuvant administered at different doses to cynomolgus macaques. This objective was achieved by combining historical data from two separate, related studies. The range of doses employed in those studies encompassed doses hypothesized to result in a varying degree of effectiveness against an EBOV challenge. The vaccine candidate under investigation used a homologous prime/boost immunization regimen based on a recombinant EBOV glycoprotein (rGP) developed by Novavax combined with a saponin-based Matrix-M^TM^ adjuvant. The vaccine candidate antigen was produced by cloning the full-length GP gene from the 2014 Makona EBOV into recombinant baculovirus and expressing it in Sf9 cells. The adjuvant consisted of two purified saponin fractions of *Quillaja saponaria* extract formulated with cholesterol and phospholipid into complexes approximately 40 nm in size [8,9]. The two animal studies were designed to provide data that would allow extrapolation from NHP efficacy and immunogenicity data to human immunogenicity data and prediction of vaccine efficacy in humans. The current statistical analysis is intended to support that objective.

Fewer and smaller outbreaks of Sudan virus (SUDV) and Marburg virus (MARV) have presented historically [10,11], making the AR and NHP correlates of protection more important for these diseases. While the specific vaccine candidates for SUDV and MARV may differ from those for EBOV, the methods used to assess immune response are similar; SUDV and MARV ELISAs are performed following the same method used for the EBOV ELISA with the exception of the reagents (e.g., coating antigen and control sera) used [12]. Thus, the statistical approach presented in this manuscript can be applied to further research for SUDV and MARV.

## 2. Materials and Methods

The presented CoP analysis utilized data from two separate, related studies. The study design, the immunological assays used for endpoint assessments, and the similarities and differences between these two animal studies are described below.

In Study 1, 24 cynomolgus macaques on study were randomized by body weight to 1 of 4 test material dose groups (5.00 µg, 2.50 µg, 1.25 µg, and 0.63 µg) of 5 or 6 animals each (2 or 3 males and 2 or 3 females) and 1 placebo control group of 2 animals (1 male and 1 female). Animals were vaccinated on Study Days 0 and 21; animals were challenged on Study Day 42. Serum was collected on Study Days 0, 7, 14, 21, and 28, though only Day 28 data were used for the presented analysis.

In Study 2, 24 cynomolgus macaques on study were randomized by body weight to two test material dose groups (10 µg and 0.63 µg) of 7 animals each (3 or 4 males and 3 or 4 females), one test material dose group of 8 animals (2.50 µg) (3 males and 5 females), and one placebo control group of 2 animals (1 male and 1 female). Animals were vaccinated on Study Days 0 and 21; animals were challenged on Study Day 42. Serum was collected on Study Days 0, 7, 14, 21, 28, and 42, though only Day 28 data were used for the presented analysis.

The overall design and conduct of the two studies were parallel. The two studies acquired the animals from the same source (Covance Research Products). Animals were in roughly the same age range (3.2–5.6 years for Study 1 and 3.5–6.4 years for Study 2). Equal numbers of males and females were used in each study. Consistent procedures were followed in each study including animal husbandry and handling, route of treatment administration (though injection volume and administration site differed), animal observations, transfer to the BSL-4 lab for challenge, and challenge administration; all activities prior to animal transfer were conducted at a single facility, and all post-transfer activities were conducted at a single BSL-4 lab. The target dose of challenge material was also the same for both studies, and actual challenge doses differed only slightly (100 PFU for Study 1 and 115 PFU for Study 2). EBOV exposures were performed with Filovirus Animal Nonclinical Group (FANG) approved stocks originating from lethal human infections [13]. Finally, the vaccination dose range was overlapping across the two studies.

Some differences existed between the two studies. For Study 1, the last specimens for PsVNA and ELISA analyses were collected on Study Day 28, while additional specimens were collected on Study Day 42 for Study 2. Because the Study Day 42 analyses were performed for only one study, the analysis in the current CoP study was limited to results for Study Day 28. End of study was defined differently for the two studies, with end of study 30–31 days post-challenge for Study 1 and 26–31 days post-challenge for Study 2. However, no animals in Study 1 succumbed between 26 and 31 days post-challenge, making this difference irrelevant. The last animal to succumb under Study 1 succumbed 14 days post-challenge and the last (only) animal to succumb under Study 2 succumbed 13 days post-challenge. Additionally, vaccine material was provided separately for each study. All doses were prepared in accordance with written instructions from the vaccine sponsor, but dose confirmation analysis was only conducted on the material prepared for Study 2. The final concentration of GP in the material prepared for administration in Study 1 was not able to be confirmed. Finally, survival outcomes were different between the two studies, with 41% (9/22) survival over all vaccine doses in Study 1 compared to 95% (21/22) survival in Study 2. At the 0.63 and 2.50 µg vaccine doses common to both studies, survival was 20% (1/5) at the 0.63 µg dose and 0% (0/6) at the 2.5 µg dose in Study 1 compared to 100% (7/7) at the 0.63 µg dose and 88% (7/8) at the 2.5 µg dose in Study 2. Differences in immune response and survival may have been due to a difference in vaccine preparation between the two studies. However, this cannot be verified, and a definitive explanation of the differences in immune response and survival cannot be provided. These differences in survival are not necessarily detrimental to demonstrating a correlation between immune response and survival.

In each study, blood samples were collected and processed into serum for evaluation of immune response induced by the immunization using the EBOV PsVNA and anti-EBOV GP IgG ELISA.

The PsVNA was used to quantify the neutralizing capability of antibodies in NHP sera against EBOV using a pseudovirion (PsV) surrogate for the virus. The PsV consisted of a replication-deficient vesicular stomatitis virus (VSV) that contained a luciferase reporter gene. This PsV, which lacks the gene encoding the VSV envelope glycoprotein, G (VSV-G), was provided an EBOV envelope glycoprotein in trans by transfecting cells with a DNA plasmid encoding the desired viral membrane protein, which was then incorporated in the PsVs. Commercially available EBOV PsVs (from IBT Bioservices) were used for conducting the EBOV PsVNA. The assay endpoints were the 50% and 80% PsV neutralization titers (PsVNT_50_ and PsVNT_80_, respectively), the reciprocals of the serum dilutions at which 50% and 80% of the input PsV was neutralized.

The NHP EBOV PsVNA was conducted per SOP. Briefly, undiluted, heat-inactivated serum was added to maintenance media (Eagle’s Minimum Essential Medium, approximately 10% heat-inactivated fetal bovine serum [HI-FBS], and approximately 1% antibiotic/antimycotic), and a 6-fold serial dilution was performed. Diluted EBOV PsV was then added to all wells creating the neutralization reaction mixture, except for the Cell Culture Control wells. The neutralization reaction mixture was then allowed to incubate for 60–75 min at room temperature (RT) and then, after removing the medium, was transferred onto VERO E6 cells seeded in black-walled opaque 96-well plates (Corning, Cat No. 3916, Corning, NY, USA). The cells with the neutralization reaction mixture were incubated at 37 ± 2 °C with 5 ± 2% CO_2_ for 60–75 min to allow PsV entry into the cells. After this incubation, pre-warmed maintenance media was added, and the plates were incubated at 37 ± 2 °C with 5 ± 2% CO_2_ for 16–26 h. Afterward, the neutralization reaction mixture was aspirated, and cells were lysed by adding 30 µL of 1X passive lysis buffer (created by diluting 5X passive lysis buffer, Promega, Cat No. E1941, into DI water, Madison, WI, United States) to each well and shaking at approximately 200 revolutions per minute (RPM) at RT for 30–45 min. Relative light unit (RLU) values were determined for the cell lysate within each well using a BioTek Synergy HTx microplate reader. The reagent injection module was set to inject 60 µL of the Working Luciferase Reagent (Promega, Cat No. E152B and E151C), pause for two seconds, and then integrate luminescence for 10 s at a gain setting of 240 for each well of the 96-well plate.

The anti-GP IgG ELISA is designed to quantify immunoglobulin subtype G (IgG) antibodies against EBOV GP using an ELISA in which purified rGP antigen is used as the solid-phase immobilized antigen and an enzyme-conjugated anti-gamma chain secondary antibody is used as the reporter or signal system. The assay endpoint is the serum mean concentration reported in ELISA units/mL. The assay was conducted as previously described [12,14], using a human serum reference standard and NHP serum negative control.

The CoP analysis was performed using data from all vaccinated animals from both studies. Immune response on Study Day 28 was used to test for a significant relationship between immune response and survival and was selected because it was the only assessment of immune response common to both studies that used samples collected after the boost immunization. Immune response was measured by:PsVNA, reported as percent neutralization in terms of:
50% (PsVNT_50_) assay titers80% (PsVNT_80_) assay titersAnti-GP IgG ELISA serum mean antibody concentration.

Logistic regression models with a fixed effect for log10 transformed immune response (PsVNT_50_, PsVNT_80_, anti-GP IgG antibodies) on Study Day 28 and a random effect for study were fitted to the data. These models were fit to determine whether there were statistically significant relationships between the immune responses and survival to end of study. The predictive ability of the model was assessed using the area under the receiver operating characteristic (ROC) curve (AUC). AUCs for each pair of the three models were compared to determine whether there were statistically significant differences using the non-parametric approach of DeLong, DeLong, and Clarke-Pearson [15]. If a statistically significant relationship between survival to end of study and any immune response was identified, then a second logistic regression model was fitted with the log10 transformed immunization dose as an additional predictor. This second model was fit to determine whether the immunization dose provided additional information or not, given the information from the immune response.

Additional descriptive exploratory analysis of the data was performed to help understand and interpret the primary results. These exploratory analyses were not determined in advance and are described in the Results Section.

## 3. Results

A summary of the survival and immune response data by dose for the two studies is presented in Table 1. For Study 1, for PsVNT_50_, PsVNT_80_, and anti-GP IgG antibodies, the immune response increases with dose from 0.63 µg to 1.25 µg to 5.00 µg with increasing survival. However, at 2.50 µg, both the immune responses (geometric means) and survival (proportions) are lower than at the 1.25 µg and 5.00 µg doses, observed in the descriptive statistics and not based on statistical comparison. For Study 2, for PsVNT_50_, PsVNT_80_, and anti-GP IgG antibodies, the immune response increases with dose. Overall, survival at 0.63 µg and 10.00 µg is greater than at 2.50 µg, though not significantly greater, observed in the descriptive statistics and not based on statistical comparison.

Mixed-effect logistic regression models were fitted to survival data from vaccinated animals as a function of the log10 transformed immune response on Study Day 28 with a random effect for study. Results of the model fitting are presented in Table 2, where the slope of the logistic model is interpreted as the expected change in log odds of survival per unit change in immune response; also shown are estimates for the model-based AUC. There is not a statistically significant relationship between either PsVNT_50_ or PsVNT_80_ and survival (*p* = 0.1069 and *p* = 0.1173, respectively). However, there is a statistically significant relationship between anti-GP IgG antibodies and survival (*p* = 0.0057). These results suggest that survival was associated with higher anti-GP IgG antibody levels, but not higher PsVNA values, in vaccinated animals.

Model AUC for all three models was greater than or equal to 0.9, suggesting good predictive ability of all three models. AUC can be interpreted as the expected proportion of survival with a higher immune response than a uniformly drawn mortality. The ROC curves for all three models are presented in Figure 1 and appear similar. To more rigorously compare the three model AUCs, differences in AUC between each pair of models were estimated and tested. All p-values were greater than 0.05, indicating that no two AUCs are statistically different from one another.

To further support the use of anti-GP IgG antibodies as a CoP, models with the log10 transformed immunization dose in addition to the log10 transformed immune response were fitted to the data. If the effect of the log10 transformed dose is not statistically significant, and the prediction model without log10 transformed dose is similar to the prediction model with log10 transformed dose, then the use of the immune response alone is supported. In this case, the dose does not significantly improve the model fit once the immune response is known, as the main effect of dose is through the immune response correlate. The model was initially fit including the interaction between log10 transformed dose and log10 transformed immune response. This interaction was not statistically significant, so the main effects model was fit. The model fitting results are presented in Table 3. Results show that the anti-GP IgG antibody concentrations continue to be statistically significant predictors of survival when dose is added to the models, while dose is not a statistically significant predictor of survival. Model AUC for the model including dose is similar to the model AUC for the model not including dose. These results further support anti-GP IgG antibody concentrations as CoP.

To understand a potential reason why the anti-GP IgG antibody concentrations were statistically significant predictors of survival while the PsVNA titers were not, though the model AUCs were comparable, assay variability was examined. Based on characterization experiments for each assay, the total assay variability is 41.3% for PsVNT_50_, 31.7% for PsVNT_80_, and 22.2% for anti-GP IgG antibodies, indicating greater variability for the PsVNA compared to the ELISA (based on observed values and not on statistical tests). For the Study Day 28 samples considered in this analysis, the geometric percent coefficients of variation (GCVs) were also calculated for each endpoint by study and survival status. The GCVs are presented in Table 4. For the two survival groups, the PsVNT GCVs appear to be greater than the ELISA GCVs (though the values were not compared statistically). Both the total assay variabilities and the percent CVs for the study data support the proposition that increased assay variability may account for non-significant model results despite the good predictive ability of the model according to the AUC.

## 4. Discussion

The primary objective of this study was to evaluate potential CoP for the Novavax EBOV GP vaccine administered at different doses to cynomolgus macaques. This objective was achieved by combining data from two separate, related studies with a combined total of 44 NHPs. In both studies, NHPs were dosed intramuscularly (IM) with the appropriate dilution of vaccine material or placebo on Study Days 0 (prime) and 21 (boost) and challenged on Study Day 42. Immune responses measured 7 days post-boost on Study Day 28 were considered as potential CoP in this evaluation. Examination of this single timepoint is a limitation of the current study; immune responses measured at other post-boost timepoints, including the pre-challenge timepoint, should be considered for future studies of EBOV, SUDV, and MARV.

Survival outcomes differed between the two studies, with 41% (9/22) survival over all vaccine doses in Study 1 compared to 95% (21/22) survival in Study 2. At the 0.63 and 2.50 µg vaccine doses common to both studies, survival was 20% (1/5) at the 0.63 µg dose and 0% (0/6) at the 2.5 µg dose in Study 1 compared to 100% (7/7) at the 0.63 µg dose and 88% (7/8) at the 2.5 µg dose in Study 2. Corresponding differences in immune response were noted (Table 1). Differences in immune response and survival may have been due to a difference in vaccine preparation between the two studies. However, this cannot be verified, and a definitive explanation of the differences in immune response and survival cannot be provided. Ideally, definitive studies would demonstrate a correlation between vaccine dose and immune response, as well as a correlation between immune response and survival outcome. In the current investigation, dose formulation analysis was not conducted on Study 1 test material, and therefore, no conclusive data are available to ensure the concentration of GP and associated adjuvant were similar for each study. We speculate that due to the differences in immune response between studies, the formulation and/or preparation of the material may have resulted in discordant concentrations and/or ratio of antigen to adjuvant. We recommend dose formulation analysis be conducted for future preclinical studies that include vaccine preparation based on the lessons learned from the execution of the current studies. This verification will allow for confidence in antigen concentration between doses and allow comparison of different vaccine doses across independent studies.

The differences in immune response and survival between studies are not necessarily detrimental to demonstrating a correlation between immune response and survival if the immune response in individual animals correlates to survival. As can be seen in Figure 2, animals with the lowest immune responses (below the ED_20_) died, while all but two animals with the highest immune responses (above the ED_80_) survived, independent of the vaccine dose received. There is substantial animal-to-animal variability in the immune response for a given vaccine dose, especially the lower vaccine doses. Despite this variability, the immune response itself correlated with survival outcome.

Logistic regression models were fitted to the survival data as a function of log10 transformed immune response. Results of the fitted models showed that anti-GP IgG antibody concentrations on Study Day 28 are statistically significant predictors of survival, with greater survival associated with greater immune response as measured by anti-GP IgG ELISA (Figure 2). The use of anti-GP IgG antibody concentrations as a CoP was further supported by examination of models that included log10 transformed dose, where dose was not a statistically significant predictor of survival. These additional models support the conclusion that all the predictive ability is in the anti-GP IgG antibody concentrations; once anti-GP IgG antibody concentrations are known, there is no information added in knowing dose. The vaccine dose was used to modulate the immune response in these studies. Despite differences in responses to the same vaccine doses between studies and among the individual animals within each study, the immune response as measured by the anti-GP IgG ELISA provided a significant CoP.

In contrast, the fitted models showed that neither PsVNT_50_ nor PsVNT_80_ are statistically significant predictors of survival. However, model prediction ability as measured by AUC showed comparability of model prediction for all three models (anti-GP IgG antibody concentration, PsVNT_50_, PsVNT_80_). The lack of statistically significant results for PsVNA results may be due in part to the greater variability in this assay compared to the anti-GP IgG ELISA. If this variability is related to assay capability, further development of the PsVNA or a replication strategy to reduce variability may be needed to support neutralizing antibodies as a CoP for EBOV vaccines. It is also plausible that the neutralizing capacity at day 7 post-boost is highly variable between NHP and a later time point might be less variable, thereby having a stronger correlation that could be detected using the logistic regression model. Modifications to the assay and animal study design could improve our understanding of EBOV vaccine COPs and could also be applied to vaccine COP studies for MARV and SUDV vaccines. While the specific vaccine candidates for SUDV and MARV may differ from those for EBOV, the methods used to assess immune response are similar. Thus, the statistical approach presented in this manuscript can be applied to further research for SUDV and MARV.

## 5. Conclusions

The current study combines data from two EBOV studies to attempt to identify potential CoP for the Novavax EBOV GP vaccine. This study was limited to investigating immune responses at 28 days (7 days post-boost; 14 days pre-challenge); future research should consider other post-boost timepoints, especially immediately prior to challenge. Additionally, future research should verify the activity of the vaccines on study, which was not conducted for both studies used in the current research, and which may have resulted in immune response differences. Despite differences in immune response and survival outcomes within groups administered similar doses from the two studies, a significant correlation between survival outcome and anti-GP IgG ELISA immune response was identified via logistic regression. This significant correlation was independent of the vaccine dose administered, further supporting its use as a CoP. Thus, we conclude the anti-GP IgG ELISA immune response may be an effective resource in the study of SUDV and MARV and development of medical countermeasures for these diseases. PsVN was not identified as a potential CoP, possibly due to high variability of the assay, which should be reduced for future studies.

## Figures and Tables

**Figure 1 vaccines-10-01338-f001:**
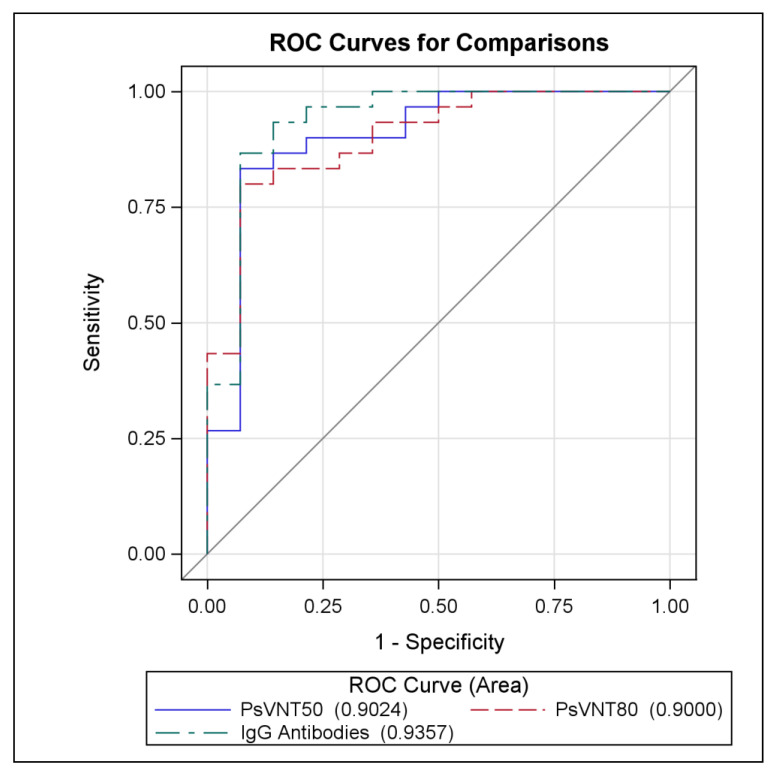
Comparison of ROC curves for the four models. ROC curves for the four models are similar.

**Figure 2 vaccines-10-01338-f002:**
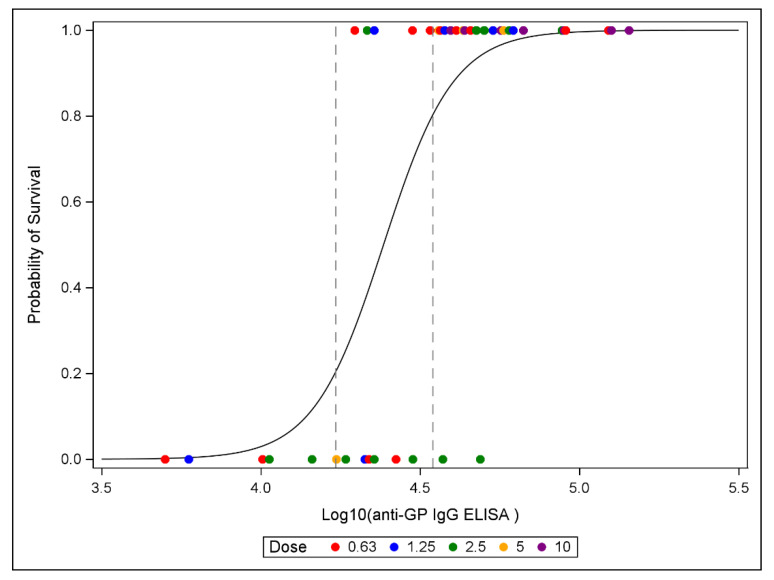
Estimated logistic model with survival data overlaid. Dashed lines represent ED_20_ and ED_80_.

**Table 1 vaccines-10-01338-t001:** Summary of survival and immune response by dose.

	Dose (µg)	N	N Survive/N	Proportion (95% Exact Confidence Interval)	Geometric Mean (95% Confidence Interval)		
					PsVNT_50_	PsVNT_80_	Anti-GP IgG (ELISA Units/mL)
Study 1	0.63	5	1/5	0.20 (0.01, 0.72)	360.15 (170.59, 760.37)	95.09 (35.44, 255.14)	16,783.67 (5730.02, 49,160.65)
	1.25	6	4/6	0.67 (0.22, 0.96)	439.01 (143.50,1343.12)	127.33 (36.49, 444.29)	26,619.14 (10,845.19, 65,335.77)
	2.50	6	0/6	0.00 (0.00, 0.46)	356.48 (247.53, 513.40)	100.40 (65.00, 155.08)	20,371.29 (12,519.54, 33,147.33)
	5.00	5	4/5	0.80 (0.28, 0.99)	617.37 (284.15, 1341.36)	153.38 (63.23, 372.04)	38,319.96 (21,021.61, 69,852.84)
Study 2	0.63	7	7/7	1.00 (0.59, 1.00)	128.03 (34.36, 477.11)	39.92 (8.78, 181.39)	44,254.96 (24,416.90, 80,210.91)
	2.50	8	7/8	0.88 (0.47, 1.00)	567.89 (372.73, 865.24)	177.48 (78.78, 399.82)	48,660.34 (35,121.68, 67,417.85)
	10.00	7	7/7	1.00 (0.59, 1.00)	808.12 (269.32, 2424.85)	220.32 (64.33, 754.52)	72,345.85 (45,475.91, 115,092.21)
Combined Studies	0.63	12	8/12	0.67 (0.35, 0.90)	197.01 (90.04, 431.07)	57.31 (23.92, 137.30)	29,547.05 (17,060.52, 51,172.40)
	1.25	6	4/6	0.67 (0.22, 0.96)	439.01 (143.50, 1343.12)	127.33 (36.49, 444.29)	26,619.14 (10,845.19, 65,335.77)
	2.50	14	7/14	0.50 (0.23, 0.77)	465.15 (350.50, 617.31)	139.03 (86.98, 222.22)	33,504.87 (23,642.36, 47,481.57)
	5.00	5	4/5	0.80 (0.28, 0.99)	617.37 (284.15, 1341.36)	153.38 (63.23, 372.04)	38,319.96 (21,021.61, 69,852.84)
	10.00	7	7/7	1.00 (0.59, 1.00)	808.12 (269.32, 2424.85)	220.32 (64.33, 754.52)	72,345.85 (45,475.91, 115,092.21)

**Table 2 vaccines-10-01338-t002:** Fitted logistic model results for survival as a function of immune response.

Immune Response	Slope Estimate	*p*-Value	Model AUC
PsVNT_50_	1.9053	0.1069	0.9024
PsVNT_80_	1.6405	0.1173	0.9000
IgG Antibodies	9.0752	0.0057 *	0.9357

* Statistically significant at the 0.05 level.

**Table 3 vaccines-10-01338-t003:** Results for models including dose.

Assay	Effect	Interaction Model		Main Effects Model		
		Estimate	*p*-Value	Estimate	*p*-Value	AUC
anti-GP IgG ELISA	Intercept	−37.1114	0.2701	−41.6292	0.2190	0.9333
	log_10_(ELISA)	8.4941	0.0300 *	9.5170	0.0070 *	
	log_10_(Dose)	−19.2246	0.6466	−0.6670	0.6538	
	log_10_(ELISA) × log_10_(Dose)	4.1043	0.6577	NA	NA	

* Statistically significant at the 0.05 level. NA indicates not applicable.

**Table 4 vaccines-10-01338-t004:** Geometric percent coefficients of variation.

Study	Survival	N	PsVNT_50_	PsVNT_80_	IgG Antibodies
1	Dead	13	61.74	70.64	66.66
1	Survive	9	67.61	93.11	31.84
2	Dead	1			
2	Survive	21	223.05	296.21	59.21

## Data Availability

All relevant data are provided in Table 1.

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
