# Peer review of "Immune Correlates of Protection from Filovirus Efficacy Studies in Non-Human Primates"

_vaccines, 2022, doi:10.3390/vaccines10081338_

Round 1

Reviewer 1 Report

D.A. Triplett, et al. present a very well executed and communicated study that identifies correlates of protection for immunization of cynomolgous macaques with Novavax Ebola Virus glycoprotein vaccine using serum from two previous studies.  The experimental approach is very well detailed and the analyses of the results are robust and thorough.  The report is timely and provides useful information for vaccine researchers in the field of Ebola Virus immunization.  Furthermore, the manuscript is clearly written so as to be easily understood even by non-experts in the field.

Minor criticisms

Abstract, Line 18:  Missing a space after the parenthetical "(PsVNA)."

Materials and Methods, Line 142:  The source of the "Commercially available EBOV PsVs" should be included in the text.

Line 145:  The word "waswas" needs to be replaced with "was."

Results:  The asterisks on numbers in Tables 2 and 3 are not explained in the tables.  I assume these are noted because they are mentioned explicitly in the text of the narrative.  If, on the other hand, the asterisks represent something with regards to statistical significance, then that should be noted.  Either way, this should be mentioned in the footer of the two tables.

Author Response

Comments and Suggestions for Authors

D.A. Triplett, et al. present a very well executed and communicated study that identifies correlates of protection for immunization of cynomolgous macaques with Novavax Ebola Virus glycoprotein vaccine using serum from two previous studies.  The experimental approach is very well detailed and the analyses of the results are robust and thorough.  The report is timely and provides useful information for vaccine researchers in the field of Ebola Virus immunization.  Furthermore, the manuscript is clearly written so as to be easily understood even by non-experts in the field.

Minor criticisms

Abstract, Line 18:  Missing a space after the parenthetical "(PsVNA)." – Space added

Materials and Methods, Line 142:  The source of the "Commercially available EBOV PsVs" should be included in the text. – Added IBT Bioservices as the source for PsVs.

Line 145:  The word "waswas" needs to be replaced with "was." – Corrected

Results:  The asterisks on numbers in Tables 2 and 3 are not explained in the tables.  I assume these are noted because they are mentioned explicitly in the text of the narrative.  If, on the other hand, the asterisks represent something with regards to statistical significance, then that should be noted.  Either way, this should be mentioned in the footer of the two tables. – Footnotes have been added to the tables to indicate the meaning of the asterisks. Asterisks indicate results are statistically significant at the 0.05 level.

Reviewer 2 Report

There are a few major problems in this manuscript.

1. Based on survival data, it shows the dose (2.5 ug) has the lower survival rates compared with 0.63 ug and 5 ug groups. However, the PVNT data and IgG data don't support the survival results at the selected timepoint. It is meaningless to do statistical analysis to find correlation between them. 

2. The PVNT and IgG data demonstrated that there may be other factors associated with survival rates. In order to try correlate these data with survival rates, the samples from other timepoints must be tested particularly those timepoints after infection. 

The current version of this manuscript does not meet the minimum standard for publishing.

Author Response

Comments and Suggestions for Authors

There are a few major problems in this manuscript.

  1. Based on survival data, it shows the dose (2.5 ug) has the lower survival rates compared with 0.63 ug and 5 ug groups. However, the PVNT data and IgG data don't support the survival results at the selected timepoint. It is meaningless to do statistical analysis to find correlation between them. – The Reviewer is correct that the survival rate is the lowest for the 2.5 mg dose group, specifically for Study 1. In Study 1, this dose group shows the lowest survival, but also lower immune response compared to the 1.25 mg dose group; immune response for the 2.5 mg dose group, especially with respect to PsVNTs, is comparable to the 0.63 mg dose group, as is the survival proportion. While the immune responses for the 2.5 mg and 1.25 mg dose groups overlap, there is a clear shift between the two groups. There is also a shift in immune response as well as survival between studies at the 2.5 mg dose. The vaccine dose is used to modulate the immune response, and it is the immune response that is correlated with survival, not the dose. Thus, the authors believe it is still meaningful to correlate the immune responses to survival. We moved discussion of the differences in immune response and survival between studies from the methods to the discussion and also revised the discussion to address this point (lines 274-297).
  2. The PVNT and IgG data demonstrated that there may be other factors associated with survival rates. In order to try correlate these data with survival rates, the samples from other timepoints must be tested particularly those timepoints after infection. – The Reviewer’s suggestion to test samples from additional timepoints is good. The authors agree that the current study is limited by the inclusion of immune response only at 28 days. The discussion was revised to reflect this weakness in the context of future research (lines 270-273).

The current version of this manuscript does not meet the minimum standard for publishing.

Reviewer 3 Report

1. The results in this manuscript are not conclusive because of the large difference in survival rates between “Study 1” and “Study 2” groups and the lack of significant difference in PsVNA assay values, which makes it impossible to determine whether this assay can be of further help in the study of SUDV and MARV. In addition, the source of the vaccine preparation for Study 1 and Study 2 should be standardized, otherwise the survival rate results obtained in this paper are too different to confirm whether Study 1 or Study 2 results are correct. I suggest that the authors should conduct another experiment with one of the studies to confirm whether the results are the same?

Other comments are shown as the below:

2. Is there a special reason why the sera collected on days 0, 7, 14, 21, 28 in Study 1 were only analyzed on day 28, and why the sera collected on days 0, 7, 14, 21, 28, 42 in Study 2 were only analyzed on day 42?

3. What are the numerical units of PsVNT50 and PsVNT80 as shown in this manuscript?

4. In the materials and methods, it was mentioned that the survival rate of Study 1 and Study 2 groups with the same concentration were very different, and the authors said that it might be due to the difference in the vaccine preparation process, but why was the difference so great?

5. Regarding the GCVs of PsVNT were larger than those of ELISA in the survival groups of Study 1 and Study 2 in this manuscript, is it possible to present statistically how many times more and whether there is a significant difference?

6. As for: “Based on experiments for each assay, the total assay variability is 41.3% for PsVNT50, 31.7% for PsVNT80, and 22.2% for anti-GP IgG antibodies, indicating greater variability for the PsVNA compared to the ELISA”, please indicate the value of the mentioned variability is defined according to which group (Study 1 or Study 2)? Please also produce a Table to present the values which obtained by this experiment.

7. What are the values of Main Effects Model without log10(ELISA) x log10(Dose) and the units of Estimate in Table 3?

8. What are the numerical units of Slope Estimate in Table 2?

Author Response

Comments and Suggestions for Authors

  1. The results in this manuscript are not conclusive because of the large difference in survival rates between “Study 1” and “Study 2” groups and the lack of significant difference in PsVNA assay values, which makes it impossible to determine whether this assay can be of further help in the study of SUDV and MARV. In addition, the source of the vaccine preparation for Study 1 and Study 2 should be standardized, otherwise the survival rate results obtained in this paper are too different to confirm whether Study 1 or Study 2 results are correct. I suggest that the authors should conduct another experiment with one of the studies to confirm whether the results are the same? – We appreciate the Reviewer’s comment. We are not able to conduct another study at this time. However, while the survival rates from the two studies at the 2.50 mg dose were different, we found that the vaccine doses were not informative when developing the correlate of protection, but that the immune response was correlated with survival. Despite differences in responses to the same vaccine doses in these studies, the immune response as measured by the anti-GP IgG ELISA provided a significant correlate of protection. Thus, the anti-EBOV GP IgG ELISA has been shown to be beneficial in the study of EBOV, indicating potential for benefit for the corresponding assays in the study of SUDV and MARV. Additional study for the PsVNA may be needed to support its benefit. We moved discussion of the differences in immune response and survival between studies from the methods to the discussion and also revised the discussion to address this point (lines 274-297).

Other comments are shown as the below:

  1. Is there a special reason why the sera collected on days 0, 7, 14, 21, 28 in Study 1 were only analyzed on day 28, and why the sera collected on days 0, 7, 14, 21, 28, 42 in Study 2 were only analyzed on day 42? – All collected samples were analyzed in the respective assays under the two studies but only results from samples collected post-boost and prior to challenge (day 28, a collection timepoint shared between studies) were considered as potential CoP. The discussion was revised to reflect this weakness in the context of future research (lines 270-273).
  2. What are the numerical units of PsVNT50 and PsVNT80 as shown in this manuscript? – PsVNT50 and PsVNT80 are unitless; these are essentially the reciprocal of the dilution that neutralizes 50% and 80%, respectively, of PsV infectivity. The manuscript states at lines 178-179 that the PsVNT50 and PsVNT80 responses are titers associated with 50% and 80% neutralization.
  3. In the materials and methods, it was mentioned that the survival rate of Study 1 and Study 2 groups with the same concentration were very different, and the authors said that it might be due to the difference in the vaccine preparation process, but why was the difference so great? -- It is unknown to the authors why the survival difference between the two studies was so great. It is thought the difference may be due to vaccine preparation (as stated in the manuscript) but this cannot be confirmed. We revised the discussion at lines 274-297 in response to this and other comments.
  4. Regarding the GCVs of PsVNT were larger than those of ELISA in the survival groups of Study 1 and Study 2 in this manuscript, is it possible to present statistically how many times more and whether there is a significant difference? – No standard statistical test for comparing GCVs exists. No changes were made to address this comment.
  5. As for: “Based on experiments for each assay, the total assay variability is 41.3% for PsVNT50, 31.7% for PsVNT80, and 22.2% for anti-GP IgG antibodies, indicating greater variability for the PsVNA compared to the ELISA”, please indicate the value of the mentioned variability is defined according to which group (Study 1 or Study 2)? Please also produce a Table to present the values which obtained by this experiment. – The values for total assay variability presented are based on historical experiments outside the scope of this manuscript and are not based on either Study 1 or Study 2. GCVs for each assay from the current study are presented in Table 4 by survival outcome. This was noted at line 250.
  6. What are the values of Main Effects Model without log10(ELISA) x log10(Dose) and the units of Estimate in Table 3? – The log10(ELISA) x log10(Dose) effect is not included in the Main Effects Model, as this term represents the interaction, so there are no values. The units of Estimates are expected log odds of survival for the Intercept term and the expected change in log odds of survival per unit change in log10(ELISA), log log10(Dose), and log10(ELISA) x log10(Dose), respectively, for the other terms in the Table 3. Units are not added to the manuscript table heading due to space limitations, but we added the interpretation of the logistic model slope at line 211-212.
  7. What are the numerical units of Slope Estimate in Table 2? – The units of the slope estimates in Table 2 are the expected change in log odds of survival per unit change in PsVNT50, PsVNT80, or IgG Antibodies concentration, respectively. Units are the standard slope units and are not added to the manuscript table heading due to space limitations, but we added the interpretation of the logistic model slope at line 211-212.

Round 2

Reviewer 2 Report

The authors provided a few sentences of answer to the questions from last review, however, without more data support, this version of manuscript will not be acceptable for publishing.

Author Response

The authors provided a few sentences of answer to the questions from last review, however, without more data support, this version of manuscript will not be acceptable for publishing.

Additional studies are not possible at this time. However, we believe that the current data support the conclusion that anti-GP IgG ELISA immune response is a correlate of protection for EBOV and can be considered for SUDV and MARV. The current data show that the highest immune responses are correlated with survival while the lowest immune responses are correlated with death, with mixed outcomes in the middle range of immune responses; see new Figure 2.

Reviewer 3 Report

I don't have any more comments based on the revised manuscript.

Once again, this manuscript do not convince me due to the  the large difference in survival rates between “Study 1” and “Study 2” groups and the lack of significant difference in PsVNA assay values, which makes it impossible to determine whether this assay can be of further help in the study of SUDV and MARV.

Author Response

I don't have any more comments based on the revised manuscript.

Once again, this manuscript do not convince me due to the  the large difference in survival rates between “Study 1” and “Study 2” groups and the lack of significant difference in PsVNA assay values, which makes it impossible to determine whether this assay can be of further help in the study of SUDV and MARV.

While there are large differences in survival between the two studies, these correlate with the large differences in anti-GP IgG ELISA immune responses between the two studies. While there is a lack of significant difference in PsVNA assay values, this measure of immune response was not found to be a significant correlate of protection. Thus, this paper concludes that ELISA may be of further help in the study of SUDV and MARV, but at this time PsVNA may not be.